# Validity and reliability of the Professionalism Assessment Scale in Turkish medical students

Esra Çınar Tanrıverdi[1], Mehmet Akif Nas[1]*, Kamber Kaşali[2], Mehmet Emin Layık[3], A. M. Abd El-Aty[4]

1 Faculty of Medicine, Department of Medical Education, Atatürk University, Erzurum, Turkey, 2 Faculty of Medicine, Department of Biostatistics, Atatürk University, Erzurum, Turkey, 3 Faculty of Medicine, Department of Medical Education, Yüzüncüyıl University, Van, Turkey, 4 Faculty of Medicine, Department of Medical Pharmacology, Ataturk University, Erzurum, Turkey

* mehmetakifnas@gmail.com

**Data Availability Statement:** All relevant data are within the paper and its Supporting Information files.

**Funding:** This study was supported by the Scientific Research Projects Coordination Unit of

## Abstract

Medical professionalism is a basic competency in medical education. This study aimed to adapt the Professionalism Assessment Scale, which is used to evaluate the professionalism attitudes of medical students, into Turkish and to assess its validity and reliability. First, the scale's translation-back-translation was performed and piloted on 30 students. Then, the final scale was applied to medical students to ensure the scale's validity. The Penn State University College of Medicine Professionalism Questionnaire was used for external validation to assess criterion validity. Confirmatory factor analysis was performed for structure validity. Test-retest, item correlations, split-half analysis, and Cronbach's alpha coefficient were evaluated to determine the scale's reliability. SPSS 25.0 and AMOS 24.0 package programs were used for statistical analysis. The statistical significance level was accepted as $P<0.05$. The mean age of the participants was 21±2 years, and 50.5% (n = 166) were female. Three hundred thirty-five students were invited, and 329 participated in the study. The response rate was 98%. The mean total Professionalism Assessment Scale score was 96.36±12.04. The three-factor structure of the scale, "empathy and humanism," "professional relationship and development," and "responsibility," was confirmed. The Cronbach's alpha coefficient of the scale was 0.94, and both the Spearman-Brown and Guttman split-half coefficients were 0.89. The three-factor structure of the scale, consisting of 22 items, explained 59.1% of the total variance. The intraclass correlation coefficient between test-retest measurements was 0.81. Confirmatory factor analysis showed a model suitable for the original version of the scale ($\chi^2$/sd = 2.814, RMSEA = 0.074). The Turkish version of the Professionalism Assessment Scale is a valid and reliable tool to determine the professionalism attitudes of medical students in Turkey.

## Introduction

Medical professionalism is the entirety of doctors' behaviors needed to earn the patients' and the public's trust while working for their benefit [1,2]. With the evolution of the physician's role from healer to professional in the last 20 years, medical professionalism has become one

Atatürk University with the project code TAB-2021-10077. The funders had no role in study design, data collection and analysis, decision to publish, or preparation of the manuscript.

**Competing interests:** The authors have declared that no competing interests exist.

of the basic competencies to be cultivated throughout medical education [3]. The concept of medical professionalism is a pledge of medical schools to society [4]. The American Board of Internal Medicine (ABIM) considers professionalism as "medicine's contract with society." The components of professionalism are grouped under six main headings: altruism, accountability, excellence, duty, honor and integrity, and respect for others [5].

The leading institutions in medical education have accepted the importance of professionalism and determined its basic principles as the priority of patient welfare, social justice, and patient autonomy. In addition to these principles, professionalism also brings liabilities such as professional competence, responsibility, patient confidentiality, respect for patients and colleagues, honesty, improving the quality of care, fair use of limited resources, managing conflicts of interest, and continuous professional development [6–8]. Professionalism is related to physician excellence, including medical knowledge, skills, and conscientious behavior [7–9]. Professional attitudes can prevent adverse medical events by providing appropriate patient care and safety [10]. In contrast, unprofessional attitudes negatively affect patient care and endanger patient safety [11].

Currently, not only patients and society but also medical associations and accreditation boards expect a physician to be professional [5,8,12]. Professionalism is one of the basic competencies accepted in pre- and postgraduate medical education in Turkey and worldwide [13].

As future professionals, medical students should reflect the public's trust in the medical profession [14]. The recognition of professionalism as a core competency required it to be integrated into medical education curricula and formally taught and evaluated. Our institution teaches professionalism to students at all stages, from their medical school entrance to graduation, with both an open and a hidden curriculum. Training and activities for professionalism begin in the first year with the white coat ceremony in the preclinical phase, the initial lessons from retired faculty members, corporate identity and physician identity lessons, continuous professional development activities, and community-based medical education practices. It continues with professional skills practices, communication skills lessons in the second year, theoretical lessons on professionalism, case analyses, vignettes and simulated/standard patient interviews in the third year. Emphasis on professional values continues during clinical years and bedside training, with a hidden curriculum. The Hippocratic Oath at graduation is also a part of this emphasis.

In competency-based medical education, it is recommended to make an evaluation to check whether students have achieved the relevant competency. Evaluation of professionalism directs learning, controls how well the objectives can be achieved and hints at the subject's importance and value [15,16]. It is recommended to use multiple and various methods to evaluate professionalism [17,18]. Evaluation methods include roleplay practices, simulated/standardized patient interviews, bedside practices and patient communication in the clinical setting, checklists, 360-degree evaluation, and various scales [19–22]. Evidence focusing on the professionalism attitudes of medical students is limited. However, a measurement tool with proven validity and reliability allows the evaluation of students' attitudes toward professionalism and the effects of time and education on these attitudes. Unfortunately, in Turkey, there is only one scale adapted to Turkish that evaluates the professional attitudes of medical students in all its dimensions. This scale is the Penn State University College of Medicine (PSCOM) Professionalism Attitude Scale [23]. Adapted to Turkish by Demirören and Öztuna (2015) [24], it was used to evaluate the professional attitudes of medical students and the effects of education [25–27].

The Professionalism Assessment Scale (PAS) is another self-report tool for assessing medical students' professionalism attitudes [28]. The PAS is a self-assessment tool developed by Klemenc-Ketis and Vrecko (2014) to measure the professionalism attitudes of medical students [28]. It has been shown that PAS covers the most basic components of professionalism,

such as empathy, humanism, professional relationships and responsibility. It has been reported that the scale is a valid and reliable tool that can be used both for the summative and formative evaluation of medical students' professional attitudes and for the self-evaluation of students [28]. The PAS is an easy-to-use tool that can be answered quickly, with fewer items than PSCOM. To the best of our knowledge, there is no Turkish or other version of the scale. This study aimed to adapt the PAS [28] tool to Turkish and conduct a validity and reliability study.

## Materials and methods

### Study design

The study is of methodological type and is a two-stage observational validation study. In the first stage, Z. Klemenc-Ketis was contacted via e-mail, and the necessary permission was obtained for the scale to be adapted into Turkish [28]. Then, the necessary permission for the study was obtained from the Ataturk University Faculty of Medicine Clinical Research Ethics Committee (No: 04/72. Date: 04.11.2021). The study was carried out under the rules of the Helsinki Declaration.

### Setting and participants

The study population consisted of Atatürk University Faculty of Medicine third-year students (n = 365). Since the theoretical courses on professionalism started in the third year of our education program, third-year students were selected as the study group. Thirty students who participated in the pilot study were excluded from the study. Thus, the study was carried out with 335 students. All students were invited to participate in the survey. Data from six students who did not complete the questionnaires were excluded from the study, and the complete data of 329 students were evaluated. Fifteen days after the first test, the same scale was sent to the students for a retest to determine the scale's reliability, and the correlation of the responses was evaluated. Seventy-five students participated in the retest. The participation rate was 98% for the first test and 22% for the retest.

Students who volunteered and gave consent were included. Data were collected through an online questionnaire prepared via Google forms. The students were informed about the purpose and scope of the study via e-mail, and the online survey link was shared in the WhatsApp class group, which included all the students. Information about the aim of the study was also included at the beginning of the questionnaire. The participants were asked to give their consent to the statement, "I voluntarily accept participation in the study". Those who did not give consent could not answer the questions. Thus, online consent from the participants was obtained.

The study was carried out between 01.12.2021 and 15.12.2021. During this period, three reminder messages were sent to the students. Identity information was not requested from the students, and data were collected anonymously. However, to match the questionnaires, the students were asked to write the last four digits of their phone numbers. The inclusion criteria for the study were determined to be being a third-year student, volunteering, and giving consent. The questionnaire took approximately 15 minutes.

### Sample size

The minimum sample size required for our study was 180 when Cronbach's alpha was 80% at the 95% confidence interval at 80% power. Considering a 10% loss, this number has been set at 200.

## Data collection tools

The data collection tool included the PAS, PSCOM Professionalism Attitude Scale Student Form (PSCOM-SF), and sociodemographic questions such as age, sex, reasons for choosing a medical school, and postgraduate plan.

**Professionalism Assessment Scale.** The PAS is a 22-item self-assessment tool that evaluates the professional attitudes of medical students [28]. The scale has a 5-point Likert scale: strongly disagree (1), disagree (2), undecided (3), agree (4), and strongly agree (5). Scale items are scored between 1–5 points. None of the items are reverse scored. The scale has three dimensions: 1) empathy and humanism (EH), 2) professional relationship and development (PR-D), and 3) responsibility (R). The total score obtained from the scale ranges between 22 and 150. Dimension scores are 10–50 points for EH, 8–80 points for PR-D, and 4–20 points for R. Higher scores indicate more positive attitudes toward professionalism. Cronbach's alpha value was determined to be 0.88.

**PSCOM Professionalism Attitude Scale-student form.** The PSCOM-SF is a scale developed by Penn State University College of Medicine (2007) to evaluate medical students' professionalism attitudes. Its Cronbach's alpha was determined to be 0.51–0.78 [23]. It was adapted into Turkish by Demirören and Öztuna (2015), and Cronbach's alpha levels of the subscales ranged from 0.46 to 0.76 [24]. There are 7 dimensions and 36 items on the scale: accountability, enrichment, equity, honor and integrity, altruism, duty, and respect. Scale items are evaluated according to a 5-point Likert system ((never (1 point). . . always (5 points))). None of the items in the scale are reverse scored. The total score obtained from the scale is between 36 and 180. Dimension scores can be calculated separately. A higher score on the scale points to a more positive attitude toward professionalism [24].

## Procedures performed within the scope of the scale's Turkish adaptation

**Language adaptation.** The translation of the scale into Turkish was made in line with the recommendations of Hilton and International Test Commission guidelines [29,30]. First, two independent experts fluent in English and Turkish were informed of Turkey's culture and were close to medical terminology due to teaching foreign languages in different medical faculties; they were asked to translate the original scale into Turkish. Before the translation, one of the translators was informed about the subject, the study's purpose, how the scale was used, and how articles about the scale were conveyed. The other translator was only asked to translate the scale. The researchers compared Turkish translations in terms of meaning and grammar, and it was determined that the translated form was not different from the original. A Turkish language expert was also consulted on the Turkish form created. Thus, the first Turkish version of the scale was obtained.

After this stage, the Turkish scale was translated back into English by an expert whose mother tongue was English and who could speak fluent Turkish, who had no knowledge of the scale and was not involved in the first translation. Researchers compared the two English versions to determine the differences between the back-translation and the original scale and found no semantic differences other than minor grammatical differences. In the translation phase, consistency in meaning was regarded instead of translating the scale items word for word. The Turkish version of the scale is presented in Table 1.

Before starting data collection, a pilot study was conducted with 30 students. The sample of the pilot application was created to be similar to the target group in terms of characteristics such as age range and sex. Participants in the pilot study were asked to read the scale items aloud and briefly explain the meaning of each. Thus, whether the students had difficulty understanding and whether there was a difference in meaning was determined. Then, the scale

**Table 1. The Turkish version of Professionalism Assessment Scale.**

| Items | |
|---|---|
| Item 1 | Hekim hasta bakarken önyargılarını bir kenara bırakmalıdır. |
| Item 2 | Hekimin mevcut kötü ruh hali hasta bakımını etkilememelidir. |
| Item 3 | Hekim hastalarıyla saygılı bir ilişki içinde olmalıdır. |
| Item 4 | Hekim iş arkadaşlarıyla saygılı bir ilişki içinde olmalıdır. |
| Item 5 | Hekim sürekli mesleki eğitim almaya devam etmelidir. |
| Item 6 | Hekim her başvuruda hastaya yardımcı olmak için elinden gelenin en iyisini yapmalıdır. |
| Item 7 | Hekim hastayı dış görünüşüne göre yargılamamalıdır. |
| Item 8 | Hekim hastanın anlayış düzeyine uyum sağlamalıdır. |
| Item 9 | Hekim hastanın istekleri için net bir sınır belirlemelidir. |
| Item 10 | Hekim öğrenciler için iyi bir rol model olmalıdır. |
| Item 11 | Hekim özel ve profesyonel yaşamı arasına net bir sınır koyabilmelidir. |
| Item 12 | Hekim ekibiyle profesyonel bir ilişki kurmayı hedeflemelidir. |
| Item 13 | İyi bir hekim olmak için çok fazla klinik bilgiye sahip olmak yeterli değildir. |
| Item 14 | Hekim-hasta iletişimi hasta yönetiminin temelidir. |
| Item 15 | Hekim, hastanın tıbbi olmayan sorunlarını da (kötü maddi durum, aile ilişkilerinde sorunlar vb.) anlamaya çalışmalı ve bunları hasta bakımına dâhil etmelidir. |
| Item 16 | Her hasta bireyselleştirilmiş bir bakımı hak eder. |
| Item 17 | Mesleki görüşünü, hastanın anlayabileceği ve kabul edebileceği şekilde hastaya sunmak hekimin görevidir. |
| Item 18 | Hekim hasta için en iyisinin ne olduğunu her zaman bilemez. |
| Item 19 | Hastanın mahremiyetini korumak hekimin yükümlülüğüdür. |
| Item 20 | Hekim hastaya ilgi göstermelidir. |
| Item 21 | Hastanın bilmediği bir şey varsa hekim bunu hastaya açıkça anlatmalıdır. |
| Item 22 | Hekimin hata yapabileceği kabul edilebilirdir. |

was given its final shape and applied to the students. Participants in the pilot study were excluded.

## Statistical analyses

SPSS v25.0 (Statistical Package for Social Sciences) and AMOS v24.0 (Analysis of a moment structure) package programs were used for the scale's validity and reliability analysis. Demographic data are given as descriptive statistics. Sociodemographic characteristics are presented as the mean ± standard deviation (SD) or as numbers and percentages. Scale scores are given as the mean±SD. Initially, in the validity analysis, Kaiser–Meyer–Olkin (KMO) and Bartlett sphericity tests were performed to evaluate whether the data were suitable for factor analysis. Then, Hotelling's $T^2$ test was used to test the differences in the mean item scores, and confirmatory factor analysis (CFA) was used to test the construct validity. Cronbach's alpha coefficient, split-half analysis, and Guttman split-half and Spearman-Brown coefficients were

analyzed for reliability. Intraclass correlation was checked with test-retest. Correlation analyses and Cronbach's alpha were used for the scale's internal consistency. Factor counts were determined by eigenvalues >1 and scree plots. The fit of the first-level CFA model results was evaluated as follows: Chi-square statistics ($\chi^2$), Chi-square degrees of freedom ratio (CMIN/DF), the goodness-of-fit index (GFI), incremental fit index (IFI), comparative index of fit (CFI), root mean square error of approximation (RMSEA), standardized root mean square (SRMR), and Tucker–Lewis index (TLI) were used. A $P$ level of <0.05 was considered significant.

## Results

Of the 335 invited students, 329 participated in the study. The response rate was 98%.

### Characteristics of participants

The participants' mean age (±SD) was 21±2 years (19–45), and 50.5% were female. The socio-demographic characteristics of the students are shown in Table 2.

### Findings regarding the validity of the scale

**Bartlett's test of sphericity and Kaiser–Meyer–Olkin measure.**   The KMO value was 0.956, and Bartlett's test of sphericity was statistically significant (Approx. = 4297.828, degrees of freedom (DF) = 231, $P$ <0.001). Thus, it was determined that the PAS scale was suitable for factor analysis. As a result of the total amount of variance explained and factor analysis of the PAS scale, it was determined that the eigenvalues of the items were grouped into three subfactors above 1.00. Of these, Factor 1 explained 48.3% of the total variance, Factor 2 explained 6%, and Factor 3 explained 4.8%. The 22-item PAS-TR explained 59.16% of the total variance. The dispersion point test determined that the scale had 3 factors, and factors after the third were not explanatory (Fig 1).

**Confirmatory factor analysis.**   According to the first level CFA model, 10 items (items 1,2,3,4,6,8,10,16,19, and 20) are collected in the "Empathy and Humanism" subdomain, 8 items (items 5,9,11,12,13,14,15,and 22) in the "Professional Relationship and Development," and 4 items (items 7, 17, 18, and 21) in the "Responsibility" subdomain (Fig 2).

Each item constituting the factors makes a statistically significant contribution to the model ($P$<0.05).

**Model fit of the scale.**   According to the goodness-of-fit analysis of the first-level CFA model, the model was compatible with the study's original structure ($\chi^2$/df = 2.81, CFI = 0.91, TLI = 0.90, RMSEA = 0.07). In our study, RMSEA, TLI, and CFI showed an acceptable fit, and SRMR showed an adequate fit. The reference values of the frequently used goodness-of-fit indexes in the literature and the goodness-of-fit analysis results are shown in Table 3.

**Table 2. Sociodemographic characteristics of participants.**

| Sociodemographic characteristics | | n | % |
|---|---|---|---|
| Gender | Female | 166 | 50.5 |
| | Male | 163 | 49.5 |
| What is your postgraduation plan? | Specialization training in Turkey | 132 | 40.1 |
| | Living abroad | 107 | 32.5 |
| | I haven't decided yet | 83 | 25.2 |
| | Working as a family doctor | 7 | 2.1 |
| Why did you choose the medical school? | My dream, my ideal and my desire to help people | 203 | 61.7 |
| | Other reasons | 122 | 37.0 |

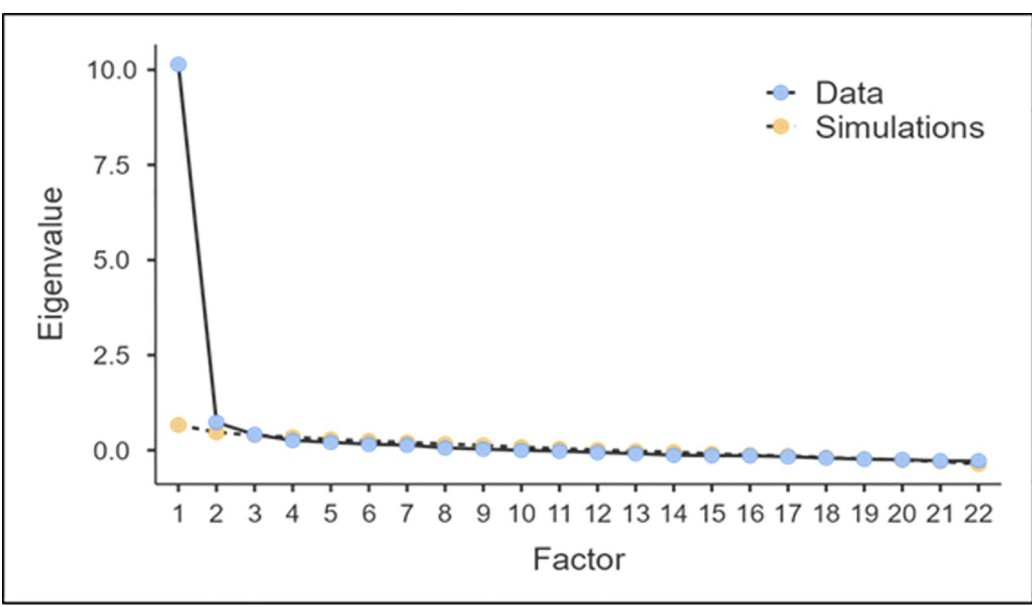

**Fig 1. Scree plot for factor analysis of the PAS-TR.**

### Findings regarding the reliability of the scale

In our study, Cronbach's alpha reliability coefficient was 0.94 for the whole scale and 0.91, 0.80, and 0.75 for EH, PR-D, and R, respectively. Cronbach's alpha values of the retest were 0.89 for the whole scale and between 0.616 and 0.847 for the dimensions. Correlations between test and retest measurements were statistically significant (Table 4).

Cronbach's alpha values were 0.88 and 0.91 in the split-half analysis performed to measure the scale's reliability. The split-half analysis findings are given in Table 5.

The analysis of variance result, performed to determine whether the scale items are additive or not, showed the scale to be additive (Nonadditivity: F = 1.193 $P$ = 0.275>0.05). A significant difference was observed in the measurement variation (between measures, F = 32.21, $P$ <0.001). The equality of the question means was tested with the Hotelling $T^2$ test, and a significant difference was found between the averages (Hotelling's T-Squared = 450.201, F = 20.131, $P$ <0.001).

### Students' PAS-TR and PSCOM-SF scores

Students' PAS-TR mean scores were above four for all items on the scale. Scale items, dimensions, and mean scores are shown in Table 6.

The PAS-TR total score was 96.36±12.04 (22–110), and the dimension scores were 44.95 ±5.80 for EH, 33.94±4.60 for PR-D, and 17.47±2.43 for R.

The PSCOM-SF total score was 155.27±16.75. There was a significant correlation between scale scores for all dimensions (Table 7).

The PAS-TR (97.59±12.12 $vs$. 94.40±11.69, $P$ = 0.001) and PSCOM-SF (158.94±14.40 $vs$. 149.32 ±18.56, $P$ <0.001) scores of students who chose medical school because it was ideal and to help others were significantly higher than those of students who chose it for other reasons. There was no significant difference between the sexes.

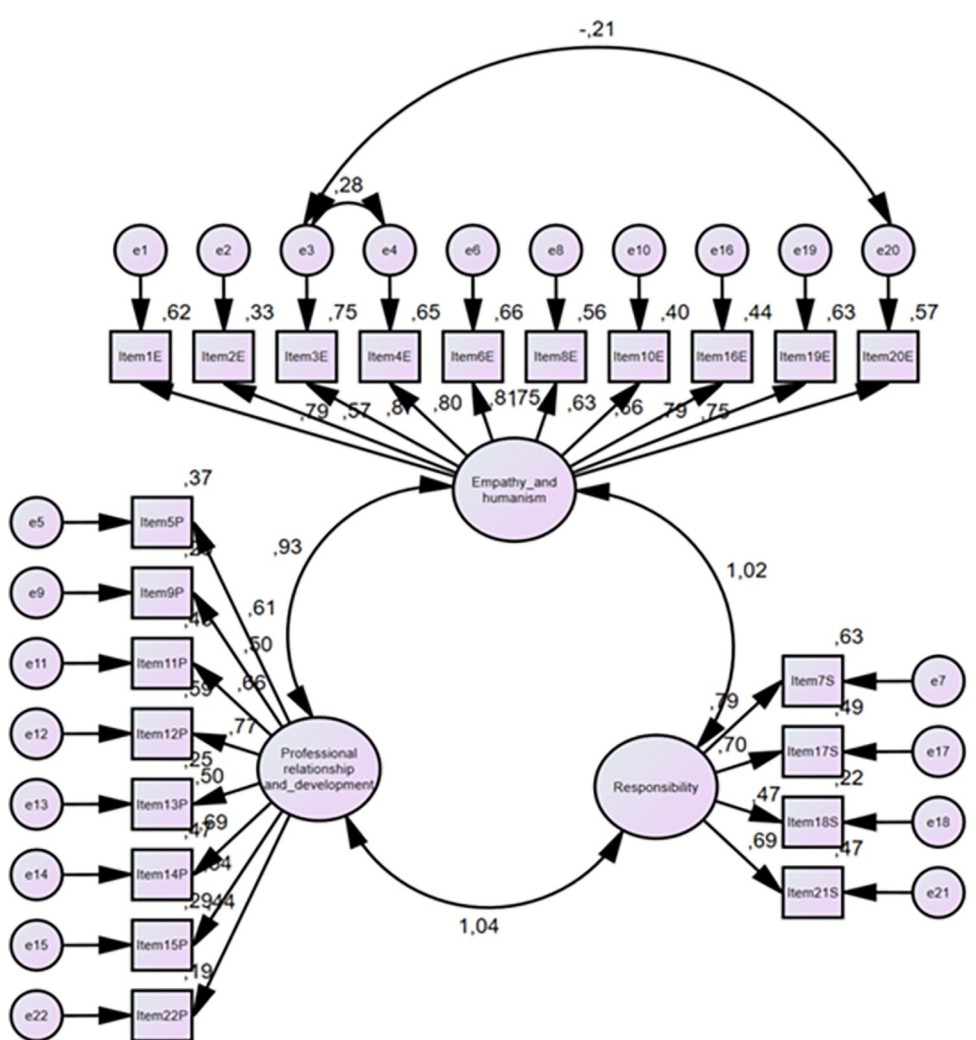

**Fig 2. Confirmatory factor analysis.**

## Discussion

Our results showed that the PAS-TR is a valid and reliable scale that can determine medical students' attitudes toward professionalism in Turkish society. In scale adaptation studies, Bartlett's test of sphericity and Kaiser–Meyer–Olkin (KMO) measurements are used to

**Table 3. Predicted goodness of fit reference values for Confirmatory Factor Analysis (CFA) and analysis.**

| Indexes Reference Value | Good fit | Acceptable fit | Measurement | Result |
|---|---|---|---|---|
| CMIN/DF | $0 < \chi 2/DF \leq 3$ | $3 < \chi 2/DF \leq 5$ | 2.814 | Good fit |
| RMSEA | $0 \leq RMSEA \leq .05$ | $.05 < RMSEA \leq .08$ | 0.074 | Acceptable fit |
| SRMR | $0 \leq SRMR \leq .05$ | $.05 < SRMR \leq .10$ | 0.040 | Good fit |
| TLI | $.95 < TLI \leq 1$ | $.90 < TLI \leq .94$ | 0.901 | Acceptable fit |
| CFI | $.95 < CFI \leq 1$ | $.90 < CFI \leq .94$ | 0.911 | Acceptable fit |

CMIN/DF: Chi-square/degree of freedom; RMSEA: Root mean square error of approximation; SRMR: Standardized root mean square residual (SRMR); CFI: Comparative fit index; TLI: Trucker-Lewis index.

**Table 4. Correlation of test and retest results.**

| | | | EH retest | PR-D retest | R retest |
|---|---|---|---|---|---|
| | | | **Correlations** | | |
| EH test | r | | .485** | .230* | .420** |
| | p | | 0.000 | 0.048 | 0.000 |
| | N | | 75 | 75 | 75 |
| PR-D test | r | | .514** | .393** | .456** |
| | p | | 0.000 | 0.000 | 0.000 |
| | N | | 75 | 75 | 75 |
| R test | r | | .433** | .347** | .418** |
| | p | | 0.000 | 0.002 | 0.000 |
| | N | | 75 | 75 | 75 |

**Correlation is significant at the 0.01 level (2-tailed)

* Correlation is significant at the 0.05 level (2-tailed)

EH, Empathy and Humanism; PR-D Professional Relationship and development.; R responsibility.

demonstrate sample size adequacy and evaluate the scale's fit for factor analysis. If the KMO value is higher than the threshold value of 0.6, Bartlett's test of sphericity should be significant [31,32]. In our study, the KMO value was 0.95, and Bartlett's test of sphericity was significant. Thus, it can be said that the scale can effectively measure the phenomenon, there is a correlation between the variables, and the study is suitable for factor analysis. Factor loads should be at least 0.30 to discuss construct validity according to sample size [33,34]. In our study, factor loading values indicate construct validity.

CFA determines the validity of measurement tools developed in other samples and cultures [35]. CFA was performed to determine whether the Turkish sample could confirm the scale's factor structure and construct validity. Various model fit indexes were used to investigate the fit of PAS-TR to the data. According to the goodness-of-fit analysis results of the first-level CFA model, the model was compatible with the original structure of the study. The DFA was first calculated by dividing the chi-square value by the degrees of freedom. A $\chi^2$/df value below 5 is considered an adequate model fit [36]. In our study, this proportion was 2.81. According to the goodness-of-fit analysis results of the first-level CFA model, the sample model was

**Table 5. Reliability statistics.**

| | | | | | |
|---|---|---|---|---|---|
| | | **Reliability Statistics** | | | |
| Cronbach's Alpha | Part 1 | Value | | | .91 |
| | | Number of Items | | | 11[a] |
| | Part 2 | Value | | | .88 |
| | | Number of Items | | | 11[b] |
| | Total N of Items | | | | 22 |
| Correlation Between Forms | | | | | .81 |
| Spearman-Brown Coefficient | Equal Length | | | | .89 |
| | Unequal Length | | | | .89 |
| Guttman Split-Half Coefficient | | | | | .89 |

[a] The items are 1, 2, 3, 4, 5, 6, 7, 8, 9, 10, and 11.
[b] The items are 12, 13, 14, 15, 16, 17, 18, 19, 20, 21, and 22.

**Table 6. Scores of PAS-TR.**

| Items | Factors | Mean Score | SD |
|---|---|---|---|
| | **Factor 1: Empathy and humanism** | **44.95** | **5.8** |
| 1 | When managing patients, the physician should put aside his/hers prejudices. | 4.56 | 0.787 |
| 2 | The current bad mood of the physician should not affect the management of patients. | 4.31 | 0.925 |
| 3 | The physician should have a respectful relationship with the patients. | 4.64 | 0.703 |
| 4 | Physicians should have a respectful relationships with their coworkers. | 4.62 | 0.661 |
| 6 | The physician should do his/her best to help the patient in every consultation. | 4.52 | 0.694 |
| 8 | The physician should adapt to the level of the patient's understanding. | 4.50 | 0.754 |
| 10 | The physician should be a good role model for students. | 4.45 | 0.803 |
| 16 | Each patient deserves individual management. | 4.29 | 0.808 |
| 19 | The physician must protect the confidentiality of the patient. | 4.60 | 0.674 |
| 20 | The physician should show interest in the patient | 4.47 | 0.777 |
| | **Factor 2: Professional relationship and development** | **33.94** | **4.6** |
| 5 | The physician should constantly engage in continuous professional education. | 4.38 | 0.796 |
| 9 | The physician should set a clear limit to which the patient can claim his/her requests. | 4.02 | 0.935 |
| 11 | The physician should be able to set a clear line between private and professional life. | 4.21 | 0.896 |
| 12 | The physician should aspire to a professional relationship in his/her team. | 4.47 | 0.737 |
| 13 | A lot of clinical knowledge is not sufficient to be a good physician. | 4.22 | 0.958 |
| 14 | Physician–patient communication is the basis of patient management. | 4.33 | 0.847 |
| 15 | The physician should also try to understand the patient's nonmedical problems (i.e., poor financial status. family relationship problems) and include them in consultation. | 4.02 | 0.970 |
| 22 | It is acceptable that the physician can make mistakes. | 4.28 | 0.944 |
| | **Factor 3: Responsibility** | **17.47** | **2.43** |
| 7 | The physician should not judge the patient by appearance. | 4.65 | 0.696 |
| 17 | It is the physician's duty to present his/hers professional opinion to the patient in such a way that the patient can understand and accept it. | 4.38 | 0.776 |
| 18 | The physician cannot always know what is best for each patient | 4.03 | 0.930 |
| 21 | The physician should tell the patient frankly if there is something he/she does not know. | 4.42 | 0.765 |

**SD** Standard deviation.

**Table 7. Correlations between PAS-TR and PSCOM-SF scale scores.**

| PAS-TR | | PSCOM-SF | | | | | | |
|---|---|---|---|---|---|---|---|---|
| | | Accountability | Enrichment | Equity | Honor and integrity | Altruism | Duty | Respect |
| EH | r | 0.568 | 0.493 | 0.522 | 0.448 | 0.483 | 0.483 | 0.361 |
| | p | 0.000 | 0.000 | 0.000 | 0.000 | 0.000 | 0.000 | 0.000 |
| PR-D | r | 0.501 | 0.403 | 0.427 | 0.393 | 0.420 | 0.466 | 0.274 |
| | p | 0.000 | 0.000 | 0.000 | 0.000 | 0.000 | 0.000 | 0.000 |
| R | r | 0.472 | 0.390 | 0.411 | 0.371 | 0.374 | 0.428 | 0.311 |
| | p | 0.000 | 0.000 | 0.000 | 0.000 | 0.000 | 0.000 | 0.000 |

**PAS-TR** Professionalism Assessment Scale Turkish Version; **PSCOM-SF** Pennsylvania State University Professionalism Attitude Scale Student Form.

consistent with the original structure of the study and was significant ($\chi^2$/sd = 2.814, RMSEA = 0.07). A CFI of 0.90 and above indicates an acceptable fit, while a CFI greater than 0.95 is considered a perfect fit [37,38]. Similarly, a TLI of 0.90 or greater is an acceptable fit, while a TLI greater than 0.95 indicates a perfect fit [39]. An RMSEA index of less than 0.08 is acceptable, and an index of less than 0.05 is considered excellent [40,41]. In our study, RMSEA, TLI, and CFI showed an acceptable fit, while CMIN/DF and SRMR showed an adequate fit.

Through factor analysis, the three-factor structure of the original scale was confirmed. While the items collected in Factor 1 were associated with the "empathy and humanism" sub-domain of the original scale, the items under Factor 2 were associated with the "professional relationship and development" dimension, and the items under Factor 3 were associated with the "responsibility" subdomain. Some items in the original scale were included in at least two factors and were placed in the most appropriate subdomain. This situation is associated with professionalism's interrelatedness and often overlapping characteristics [28]. In the current study, since all items were under the same factors as the original structure, the names of the dimensions were kept and formatted in parallel with the original scale [28].

The variance explained in the scales should be at least 50% of the total variance; representativeness cannot be asserted if it explains any less [35]. In our study, the PAS-TR explained 59.1% of the total variance. The higher the explained variance is, the better a concept or construct is measured [35]. In the original scale, the full scale explained 46.8% of the variance [28].

Empathy and humanism are core values of medical professionalism [28,42]. As in the original scale, empathy and humanism made up the majority of the variance in our study and were revealed as the main factors. The "professional relations and development" dimension is the second most crucial component of the variance. Since this subdomain covers continuing professional development, lower grades from undergraduate students could be expected, but our study did not confirm this. Our results are consistent with the literature [43,44]. Responsibility, the third dimension of the scale, is a vital component of professionalism. While providing health services, a physician is responsible for the patient, society, and profession.

Intraclass correlation showing temporal consistency and reliability was significant in test-retest measurements. This result shows that the scale measurements are consistent in a certain period. Furthermore, in the test and retest, the internal consistency of the scale items and dimensions was appropriate.

According to the Cronbach's alpha values obtained, no item was required to be removed if an item was deleted from the scale. When assessing the scale's internal consistency, it is recommended to calculate Cronbach's alpha values for each dimension and the overall scale [45]. A Cronbach's alpha value of at least 0.70 is recommended for acceptable internal consistency [35]. In the current study, the scale's Cronbach's alpha values were 0.94 for the whole scale and between 0.75–0.80 for the subdomains. The internal consistency of the original scale dimensions was between 0.60–0.84 [28]. According to all these findings, we can state that the whole scale and its domains are reliable. In various studies using different professionalism scales, internal consistencies ranged between 0.71 and 0.86 [43,46,47].

Examining the studies using the PAS scale in the literature, the total score was 90.9±8.9 in the study of Klemenc-Ketis & Vrecko and 92.6±6.1 in the study of Selic et al. [28,46]. In our study, attitude scores were higher than those in these studies. This suggests that our students are aware of medical professionalism and have a positive attitude toward professionalism.

Various studies have reported that women's professionalism attitude scores are higher than men's [24,26,27,47]. However, in the current study, no significant difference was found between the sexes.

Empathy was reported as the main factor in the original scale, as in our study, and this result was associated with the high female population in the study sample [28]. However, since our study's male/female ratio was similar, we cannot discuss such a relationship. In addition to the differences in medical school curricula in countries where the studies were conducted, the culture and environment may also have influenced these differences.

Our study used the PSCOM-SF as an external scale for measuring professionalism attitudes to evaluate the scale's convergent validity. A positive and significant correlation was found between the scores of both scales. This result shows that both scales measure the same characteristics. A previous study reported that the attitude scores of students who prefer medical school because it is ideal and to help people are higher than those of students who choose it for other reasons [26]. Our study confirmed this result with both the PAS-TR and PSCOM-SF scales. This finding suggests that making conscious choices significantly impacts students' professional attitudes.

The results showed that the PAS-TR is a valid and reliable scale for evaluating the professionalism attitudes of medical students. It was observed that the internal consistency of the PAS-TR was high, and it provided criterion validity. The scale covers the main factors related to medical professionalism. Psychometrically, the three-dimensional structure of the scale was confirmed with adequate fit values. This shows that the scale is a good measurement tool for determining the professionalism attitudes of medical students. The scale can be used to evaluate medical students' professional attitudes, in formative assessments, in determining the effects of time and education on professional attitudes, and in students' self-evaluation.

## Limitations and challenges in using the Professionalism Assessment Scale in medical students

There are several potential limitations and challenges in using the Professionalism Assessment Scale (PAS) in medical students. Some of these may include the following:

Limited research: PAS has not been extensively studied, particularly in the context of medical education. More research is needed to fully understand the scale's reliability and validity in this population.

Subjectivity: The PAS relies on ratings of professionalism by individuals, which can be subjective. Different raters may have different perceptions of what constitutes professionalism, which could affect the scale results.

Cultural differences: The PAS may not be fully applicable to medical students from other cultural backgrounds. It may be necessary to adapt the scale or develop a new measure to assess professionalism in these students.

Respondent burden: The PAS is a long scale with many items, which may be burdensome for respondents to complete. This could affect the reliability and validity of the results.

Limited focus on specific domains of professionalism: The PAS assesses several domains of professionalism, but it may not capture all aspects of this complex construct.

## Study strengths and limitations

The study's strength is that it provides researchers with an instrument with proven validity and reliability, as well as international comparability of results. And the PAS is a short scale compared to other scales that can be used to measure professionalism in Turkish. However, the following limitations should be acknowledged as well. It is a cross-sectional study that was conducted on third-year students of a single medical school. The results may not represent all medical students, and their generalizability is limited. Another limitation is the low number of students who participated in the retest.

## Conclusions

It has been proven that the PAS-TR scale is valid and reliable in measuring the professionalism attitude of medical students. During the validation process, the 22-item, three-factor structure of the original version of the scale was preserved. Therefore, the scale can be used as a practical scale that can be answered in a short time, with a small number of items, in evaluating the professionalism attitudes of medical students. Further studies are needed to determine the use of the scale in medical residents and physicians. It would be beneficial to test the scale in various health professional groups and larger samples. It must be noted that the validity and reliability of the PAS may vary depending on the specific population being studied and the context in which the scale is being used.

## Supporting information

**S1 File. Professionalism Assessment Scale and the Turkish translation.**
(DOCX)

**S1 Dataset.**
(SAV)

## Acknowledgments

We thank Professor Z. Klemenc-Ketis for her cooperation in validating the Professionalism Assessment Scale in Turkish medical students. In addition, we thank the students who participated in the study.

## Author Contributions

**Conceptualization:** Esra Çınar Tanrıverdi, Mehmet Akif Nas, Kamber Kaşali, A. M. Abd El-Aty.

**Data curation:** Esra Çınar Tanrıverdi, Mehmet Akif Nas, Kamber Kaşali.

**Formal analysis:** Kamber Kaşali.

**Funding acquisition:** Esra Çınar Tanrıverdi, Mehmet Akif Nas, Kamber Kaşali.

**Methodology:** Esra Çınar Tanrıverdi, Mehmet Akif Nas, Kamber Kaşali.

**Project administration:** Esra Çınar Tanrıverdi, Mehmet Akif Nas.

**Supervision:** Mehmet Emin Layık, A. M. Abd El-Aty.

**Visualization:** A. M. Abd El-Aty.

**Writing – original draft:** Esra Çınar Tanrıverdi, Mehmet Akif Nas.

**Writing – review & editing:** Esra Çınar Tanrıverdi, Mehmet Akif Nas, Kamber Kaşali, Mehmet Emin Layık, A. M. Abd El-Aty.

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
