## [Decision Letter · Decision Letter 0]

2 Jan 2023

PONE-D-22-21345Validity and reliability of the Professionalism Assessment Scale in Turkish medical studentsPLOS ONE

Dear Dr. Nas,

Thank you for submitting your manuscript to PLOS ONE. After careful consideration, we feel that it has merit but does not fully meet PLOS ONE’s publication criteria as it currently stands. Therefore, we invite you to submit a revised version of the manuscript that addresses the points raised during the review process.

We look forward to receiving your revised manuscript.

Kind regards,

Frantisek Sudzina

Academic Editor

PLOS ONE

and https://journals.plos.org/plosone/s/file?id=ba62/PLOSOne_formatting_sample_title_authors_affiliations.pdf.

Reviewers' comments:

Reviewer's Responses to Questions

**Comments to the Author**

1. Is the manuscript technically sound, and do the data support the conclusions?

Reviewer #1: Yes

Reviewer #2: Yes

2. Has the statistical analysis been performed appropriately and rigorously? 

Reviewer #1: Yes

Reviewer #2: Yes

3. Have the authors made all data underlying the findings in their manuscript fully available?

Reviewer #1: No

Reviewer #2: No

4. Is the manuscript presented in an intelligible fashion and written in standard English?

Reviewer #1: Yes

Reviewer #2: Yes

5. Review Comments to the Author

Reviewer #1: Abstract: please, add the number of invited participants and the number of participants that participated in the study.

Introduction: very good

Methods:

Methods are not well structured. For example, under the Participants, you report also the data collection procedure. Please, be consistent.

I am not sure about the sampling procedure. Did you invite all students? How did you invite them?

Line 117: please, delete xxx.

Results:

At the beggining, please, state how many students participated in the study and what was the response rate.

Lines 236-238: you are repeating here the methods. Similarly, also in some other places in text under Results, you are repeating the methods. This should be deleted.

Discussion: very good

Reviewer #2: Dear authors,

Thank you for your efforts invested in preparing the manuscript. I would like to recommend the following points to take into consideration for the improvement of your manuscript.

1. In the introduction, it might be better to include more information about the Professionalism Assessment Scale (PAS), if it has been translated and validated to other languages, etc.

2. In the method, the respondents were recruited from the third year. Is there any specific reason for selecting the third-year students for this study?

3. Please describe how the re-test 75 respondents were selected from the total of 329 respondents.

Thank you.

6. PLOS authors have the option to publish the peer review history of their article (what does this mean?). If published, this will include your full peer review and any attached files.

Reviewer #1: No

Reviewer #2: No

---

## [Author Response · Author response to Decision Letter 0]

11 Jan 2023

PONE-D-22-21345

Response to Reviewers

Dear Sudzina and reviewers,

Thank you for giving us the opportunity to submit a revised draft of the manuscript “Validity and reliability of the Professionalism Assessment Scale in Turkish medical students” for publication in the “PLOS ONE”. We want to thank you and the reviewers for their careful and positive evaluation of our manuscript. We also appreciate the time and effort the reviewers have dedicated to providing their valuable feedback and insightful comments on our paper. Below, you can find a point-by-point response to the suggestions made by the reviewers. Changes made in the manuscript are marked “red.”

Yours truly,

The corresponding author on behalf of the authors.

Response to reviewer # 1 comments:

 Reviewer 1 Comment Author response

1. Abstract: please, add the number of invited participants and the number of participants that participated in the study.

Author Response: The number of students invited and participated in the study and the questionnaire response rate were added to the abstract.

2. Methods are not well structured. For example, under the Participants, you report also. Please, be consistent.

Author Response: The method section has been restructured.

3. I am not sure about the sampling procedure.

1. Did you invite all students?

2. How did you invite them?

Author Response:1. All students (n=365) were invited to participate in the study. Thirty students participating in the pilot study were excluded, and the study was conducted with the remaining 335 students. Three hundred twenty-nine students volunteered to participate. The participation rate was 98%.

2. The students were informed via e-mail, and the survey link was shared in the WhatsApp group of the third graders. All of the students were included in the WhatsApp group.

Line 117: Please, delete xxx. The sampling procedure was explained in detail in the "Setting and participants" section.

Author Response: XXX was written for blinding purposes. It was changed to Atatürk.

4. Results: At the beginning, please, state how many students participated in the study and what was the response rate.

Author Response: For the retest, the questionnaire was resubmitted to the same group of students (n=335), and all were included. However, only 75 students answered the survey (added to the Settings and participants section).

The small number of students participating in the retest was stated as a limitation of the study.

Lines 236-238: you are repeating here the methods. Similarly, in some other places in the text under Results, you repeat the methods. This should be deleted.

Author Response: Duplicates on lines 236-238 have been deleted. The findings section was reviewed, and the duplicates were deleted. 

Response to reviewer # 2 comments

 Reviewer 2 Comment

1. In the introduction, it might be better to include more information about the Professionalism Assessment Scale (PAS), if it has been translated and validated into other languages, etc.

 Author response: The authors could find no other version of the PAS tool. (We also asked Prof. Dr. Klemenç, who developed the scale, but we could not find any information about its adaptation). We added this information and some other informations to the introduction of the article.

2. In the method, the respondents were recruited from the third year. Is there any specific reason for selecting third-year students for this study?

 Author response: In our education program, theoretical training and an open curriculum for professionalism start in the third year. For this reason, third-year students were chosen as the sample (added to the method section).

3. Please describe how the retest of 75 respondents were selected from the total of 329 respondents. 

 Author response: For the retest, the questionnaire was resubmitted to the same group of students (n=335), and all were included. However, only 75 students answered the survey (added to the Settings and participants section). The small number of students participating in the retest was stated as a limitation of the study.

---

## [Editor Report · Decision Letter 1]

13 Jan 2023

Validity and reliability of the Professionalism Assessment Scale in Turkish medical students

PONE-D-22-21345R1

Dear Dr. Nas,

We’re pleased to inform you that your manuscript has been judged scientifically suitable for publication and will be formally accepted for publication once it meets all outstanding technical requirements.

Kind regards,

Frantisek Sudzina

Academic Editor

PLOS ONE
---

## [Editor Report · Acceptance letter]

17 Jan 2023

PONE-D-22-21345R1 

Validity and reliability of the Professionalism Assessment Scale in Turkish medical students 

Dear Dr. Nas:

I'm pleased to inform you that your manuscript has been deemed suitable for publication in PLOS ONE. Congratulations! Your manuscript is now with our production department. 

Kind regards, 

on behalf of

Dr. Frantisek Sudzina 

Academic Editor

PLOS ONE